# Study on Microstructural and Mechanical Properties of an Al–Cu–Sn Alloy Wall Deposited by Double-Wire Arc Additive Manufacturing Process

**DOI:** 10.3390/ma13010073

**Published:** 2019-12-22

**Authors:** Shuai Wang, Huimin Gu, Wei Wang, Chengde Li, Lingling Ren, Zhenbiao Wang, Yuchun Zhai, Peihua Ma

**Affiliations:** 1College of Metallurgy, Northeastern University, Shenyang 110000, China; wangshuai106123@126.com (S.W.); lichengde20031698@126.com (C.L.); renlingling27@126.com (L.R.); zhaiyc@smm.neu.edu.cn (Y.Z.); mapeihuadbdx@126.com (P.M.); 2Inner Mongolia Metal Material Research Institute, Baotou 014000, China; wwneu@hotmail.com; 3Fushun Donggong Metallurgy & Materials Technology Co., Ltd., Fushun 113000, China; zhenbiaowang@hotmail.com

**Keywords:** Al–Cu–Sn alloy, double-wire + arc additive manufacturing, microstructure, mechanical property

## Abstract

In this present study, single-wire and double-wire Al–Cu–Sn alloy walls were fabricated by an arc additive manufacturing process. The surface morphology, elemental composition, and microstructure were investigated by scanning electron microscopy (SEM), energy dispersive spectroscopy (EDS), and transmission electron microscopy (TEM) techniques. The mechanical properties of both the single-wire and double-wire walls were studied by mechanical property testing. The results showed that the heat input of the double-wire wall was lower than that of the single-wire wall at the same wire feeding speed. The surface microstructure of the double-wire wall showed a more uniform surface than the single-wire wall. The grains of the double-wire wall were found to be isometric crystals in the as-deposited state. The *θ* phase of the double-wire wall was dispersed with a smaller grain size in the grain boundary. After T6 heat treatment, the *θ* phase of the double-wire wall was completely dissolved into the aluminum matrix, and a large amount of θ**^’^** enhanced phases were precipitated with a phase spacing of about 15 nm. The mechanical properties of the double-wire wall were shown to have significantly improved performance, which further increased to 490 MPa, 420 MPa, and 12%, respectively. The transverse and longitudinal mechanical properties of the double-wire wall were consistent, and the fracture mode of both was ductile fracture.

## 1. Introduction

Al–Cu alloys are widely used in aviation and aerospace fields as they offer excellent mechanical properties compared to conventional alloys [1,2,3]. In recent years, the use of Al–Cu alloy as a raw material to make products through a wire arc additive manufacturing (WAAM) process has become a research hotspot. Cong et al. [4] studied the effect of cold metal transfer (CMT) on the pores of an Al–Cu alloy wall. Gu et al. [5] researched the microstructure and properties of 2319 alloy walls manufactured using the CMT process enhanced by interlayer rolling. Wang et al. [6] investigated the structure and properties of a ZL205A alloy wall produced by the WAAM process. Qi et al. [7] investigated the effect of solid solution temperature on the performance of a 2024 alloy wall. These studies showed that the wall of Al–Cu alloy has excellent microstructural properties and has broad prospects for industrial application. However, all the above studies formed walls through a single-wire process without considering the influence of heat input on the structure and performance. Heat input was one of the important factors affecting the forming and performance of WAAM walls.

CMT Twin is a welding process developed by the Fronius company (Pettenbach, Austria), which combines CMT and TIME Twin processes. It has the characteristics of low heat input, high cladding efficiency, mutual heat support between front arc and back arc, and good arc stability [8]. Liu et al. [9] used a CMT Twin process to weld high-strength steel, and the performance of welded joint was good. Han et al. [10] used CMT Twin technology to weld 1561 aluminum alloy, which effectively controlled the problem of heat-affected zones and joint softening. At present, the majority application of the CMT Twin process in aluminum alloy additive manufacturing is deploying alloy composition [5,7], but the effect of this process on the microstructure and properties of walls, and the research on production and application, has not yet been reported.

In this paper, Al–Cu–Sn alloy is used as the raw material, in which Sn can refine the grain of wall and promote the precipitation of the *θ***^’^** phase and keep it stable during the aging process. The CMT Twin process was adopted for deposition, and the appearance and organizational properties of the wall were examined. A comparison with the single-wire CMT was conducted, which laid a foundation for the industrial application of CMT Twin technology of Al–Cu alloy produced by WAAM.

## 2. Experimental

### Materials and Methods

The Al–Cu–Sn alloy welding wire used in this experiment was produced by North East Industrial Materials & Metallurgy Co. Ltd. (Fushun, China), and had a diameter of 1.2 mm. The chemical composition of the main alloy elements and impurities of the welding wire are shown in Table 1. A 10 mm 2219 aluminum plate, purchased from Northeast Light Alloy Co. Ltd. (Harbin, China), was used as the base plate in the additive process.

The additive manufacturing process used two Fronius Advance 4000 arc welding power supplies and an ABB 2600 welding robot. Schematic diagrams of double-wire and single-wire process are shown in Figure 1a,b. The additive manufacturing process is shown in Figure 1c. The *x*-axis corresponds to the front of the wall, the *y*-axis corresponds to the heat source movement direction, horizontal along the wall, and the *z*-axis corresponds to the direction of growth, perpendicular to the wall.

The print parameters are shown in Table 2 where *I* is the welding current, *U* is the welding voltage, *ν_WFS_* is the wire feeding speed, and *ν_TS_* is the torch traveling speed. The wall was heat-treated at a solution temperature of 535 °C for 600 min with a quenching water temperature of 40 °C and aging at 175 °C for 240 min.

The chemical composition of the wall was tested using an ICAP7400 plasma spectrometer (Thermo Scientific, Waltham, MA, USA), while the mechanical properties were tested using a WDW-300 micro-controlled electronic universal testing machine (Changchunkexin, Changchun, China). The microstructure and morphology were observed using a metallographic microscope (OM) and a scanning electron microscope (SEM), and elemental and phase analyses are conducted by energy dispersive spectroscopy (EDS). The morphology of the precipitated phase was observed using transmission electron microscopy (TEM). The sampling position and shape of the mechanical sample are shown in Figure 1b. Tensile samples with an amount of 4 are captured at positions 1 and 2, and metallographic and transmission samples are captured at position 3. The tensile sample is fabricated into a plate with a standard distance of 30 mm and a cross-sectional area of 2.5 × 10 mm^2^.

## 3. Results and Discussion

### 3.1. Surface Morphology of Walls

The surface morphology of single-wire and double-wire walls are shown in Figure 2. It can be seen from this figure that the surface of both processes present a silver-white metallic luster and periodic concave and convex contour caused by layer-by-layer accumulation. The contour of the double-wire wall is more uniform, and the surface is more even. This is because the interaction between the two arcs of the double-wire make the arc more stable and the molten pool formed is more uniform. The layer height of each single-wire wall is 1.2 ± 0.1 mm with a thickness of 10 ± 0.5 mm, while the layer height of each double-wire wall is 1.6 ± 0.1 mm with a thickness of 7.5 ± 0.3 mm. This is because the single wire was divided into two wires after wire feeding; the heat input of each wire was greatly reduced and the size of the weld pool formed reduced in the same way. Therefore, for a wall with the same target thickness, the total feeding speed of the double-wire is higher than that of the single-wire, so the stacking speed of the double-wire is faster, and the efficiency is higher. For large size structural parts, double-wire printing efficiency can reach 3 kg/h, greatly reducing the production cycle.

### 3.2. Porosity of Walls

Porosity can cause stress concentration and reduce the effective stressed area, which has a significant impact on the performance of aluminum alloy [11,12]. Figure 3 shows the pore distribution of single-wire and double-wire walls. It can be seen from Figure 3 that the single-wire wall has a large number of pores sized between 50 and 100 microns, which are mainly distributed between layers (Figure 3a). This phenomenon is consistent with the report of Cong et al. [4]. The number of pores in the double-wire wall decreased and the approximate diameter is 50 microns. The heat input of the two processes can be calculated from Equation (1) [13]:*HI* = *ηUI*/*v_TS_*(1)

In the equation, *U* is the average voltage applied during the accumulation process, *I* is the average current applied during accumulation process, and *V_TS_* is the welding speed. The *η* has a constant value of 0.8 in the energy utilization [14]. *HI* values are calculated to be 143.54 and 109.0 J/mm for the single wire and double wire, respectively. The heat input of the double wire is much smaller than that of the single wire, so the superheat degree of the double-wire pool is much less than that of the single-wire. The pores in the welded aluminum alloys are mainly formed by hydrogen, and are formed because of the difference in the solubility of hydrogen between solid and liquid states [15]. The solubility of hydrogen in the aluminum solution increases with an increase of temperature [16], so the greater the heat input is, the more dissolved hydrogen there will be in the molten pool. The formation of pores goes through the nucleation stage first, and the nucleation velocity can be obtained from Equation (2) [17]:(2)$$j\;=\;Ce^{-\frac{4\pi r\sigma }{3KT}}$$
where *j* represents the number of nucleation per unit time, *r* represents the critical radius of the bubble, *K* represents the Boltzmann constant (*K* = 1.38 × 10^−16^ erg/K), and *σ* represents the surface tension. According to Equation (2), the nucleation rate will increase with increasing temperature. Each layer height of the single-wire wall is 1.2 ± 0.1 mm, while each layer height produced by each single welding wire in the double-wire wall is 0.8 ± 0.05 mm, so the overflow channel of pores is shorter. In conclusion, the number of pores in the double-wire wall is much lower than that of the single-wire wall.

### 3.3. Microstructure of Walls

#### 3.3.1. Microstructure of Walls in As-Deposited State

The microstructure of single-wire wall and double-wire wall in as-deposited state is shown in Figure 4. As can be seen from Figure 4a, the grains of the single-wire wall are mainly isometric crystals with a small amount of columnar crystals mixed together. The direction length of the columnar crystals is parallel to the growth direction of the wall, which is determined by the temperature gradient. The grains of the double-wire wall are all isometric crystals, as shown in Figure 4b. The grain size is significantly reduced and evenness is improved. The difference in the microstructure between the single-wire wall and double-wire wall is due to the fact that the heat input of the double-wire process is only half that of the single-wire under the same amount of wire feeding. As a result, the superheat degree of the molten pool is smaller for the double-wire wall, the solidification speed is faster, and small isometric crystals are formed. The types of precipitated phases in the as-deposited states of single-wire and double-wire walls were consistent, with *T* phase and *θ* phase as the main ones, as shown in Figure 4c,d. Compared to the single-wire wall, the precipitated phase of the double-wire wall is more evenly distributed in the grains or on the grain boundary. This is due to the relatively slow solidification speed of the melting pool in the single-wire process and that the solubility of Cu in liquid aluminum is much higher than that in the solid state, so the content of Cu in the later solidified aluminum liquid is higher and the size of the precipitated *θ* phase is larger with little segregation. The solidification speed of the double-wire wall is quick, and the solidification process of the molten pool has no particular order, so the precipitated phase is evenly distributed.

#### 3.3.2. Microstructure of Walls in T6 State

The microstructure of the single-wire and double-wire walls after T6 treatment are shown in Figure 5. As can be seen, the grains of both the single-wire and double-wire walls are isometric crystals due to the conditioning effect during the heat treatment process. The grain size of the double-wire wall is still lower than that of the single-wire wall after heat treatment due to the genetic effect of the alloy. In the microstructure of the single-wire wall, as shown in Figure 5a, besides the small black phases (A_1_) there are gray phases of a larger size distributed on the grain boundary (A_2_). The EDS energy spectrum showed that the small black phases were the re-melted *T* phase after heat treatment, as shown by C_1_ in Figure 5c, while the gray phases were *θ* phase, as shown by C_2_ in Figure 5c; the remaining *θ* phases were not completely dissolved in the aluminum matrix during heat treatment. In the microstructure of the double-wire wall after heat treatment, almost all are small-size black re-melted *T* phase, as shown in Figure 5b B_1_ and Figure 5d D_1_, indicating that the *θ* phases had been completely dissolved in the aluminum matrix during solid solution. Because the size of the *θ* phase in the double-wire wall in the as-deposited state is smaller, the contact area with the Al matrix is relatively larger, and it is thus dissolves more easily into the Al matrix during solid solution treatment. The amount of Cu dissolved in the matrix directly determines the density, size, and phase spacing of the main enhanced phase in the aging process [18]. As intermetallic compounds, the remainder of the large *θ* phases of the single-wire wall are the initiation position of the fracture, which have a negative effect on the mechanical properties.

#### 3.3.3. Strengthening Phase at Peak Aging

The morphology of the enhanced phases in the single-wire and double-wire walls under peak aging state are shown in Figure 6. On the <001> band axis, both walls show three morphologies of the precipitates, all of which are the *θ**’* phases [19]. The *θ**’* phase is the main strengthening phase in the Al–Cu alloy precipitated during the aging process. In a single-wire wall, the *θ’* phase is oval plate shaped with a length of 150 nm, a width of 100 nm, and a thickness of 7 nm. Furthermore, these distributions are sparse, and the phase spaces are 40 nm. The size of the *θ**’* phase in the double-wire wall decreases significantly, with a length of 100 nm, a width of 70 nm, and a thickness of 5 nm. The phase spaces are roughly 15 nm. The increase in the number of *θ**’* phases in the double-wire wall is mainly due to the fact that the *θ* phase is completely dissolved in the A_l_ matrix during the solid solution process, while there are still remaining *θ* phases in the single-wire wall. The ageing precipitation process of the Al–Cu alloy is [20]: supersaturated solid solution (α ss)–GP zone (GP I area)–*θ**’’* (GP II area)–*θ****^’^***–*θ*. The copper atoms diverged to form the GP region and, thus, the *θ**’* phase was formed. Therefore, the higher the number of copper atoms dissolved in the A_l_ matrix, the higher number of *θ**’* phases that can precipitate out.

### 3.4. Mechanical Properties and Fracture Morphology of Walls

#### 3.4.1. Mechanical Properties of Walls

The mechanical properties of single-wire and double-wire walls after heat treatment are shown in Figure 7. Here, 1 and 2 represents the transverse mechanical and longitudinal mechanical properties of the single-wire wall, while 3 and 4 represent the transverse mechanical and longitudinal mechanical properties of the double-wire wall. It can be seen from the Figure 7 that the mechanical properties of the single-wire wall are low, and there are differences between transverse and longitudinal properties, especially in elongation. This is because a large number of pores parallel to the accumulation layer are distributed in the wall, as shown in Figure 3. These pores have a great influence on the longitudinal mechanical properties, especially elongation. Compared to a single-wire wall, the mechanical properties of the double-wire wall have greatly improved tensile strength (490 Mpa), yield strength (420 Mpa), and elongation (12%). The mechanical properties of horizonal direction and vertical direction are consistent. The improvement in the mechanical properties is, first, due to the lower heat input of the double-wire process, which brings about the effect of fine grain strengthening. Second, the *θ* phases in the as-deposited state of the double-wire wall are smaller in size and distributed in the dispersion, and can be completely dissolved into the matrix. In the aging process, more strengthened *θ**’* phases precipitate out and the role of precipitation strengthens. Finally, the internal defects of the double-wire wall are lower and, as a result, the longitudinal mechanical properties, especially the longitudinal elongation, are greatly improved.

#### 3.4.2. Fracture Morphology of Walls

The fracture morphologies of single-wire and double-wire walls are shown in Figure 8, where (a) and (b), respectively, represent the longitudinal and transverse fracture morphologies of single-wire walls, and (c) and (d) represent the longitudinal and transverse fracture morphologies of double-wire walls. It can be seen from Figure 8 that the morphology of longitudinal fractures in the single-wire wall includes a large number of cleavage planes with only a few shallow dimples, indicating that the fracture mode is an intercrystalline brittle fracture. The transverse fracture of the single-wire wall is composed of cleavage planes and dimples, which indicates that it is a mixed-fracture mechanism. This is mainly because, after heat treatment, there are unsolved *θ* phases on the grain boundary which affect the toughness of the alloy as the initiation source during fracture. Both longitudinal and transverse fractures of the double-wire wall are filled with a large number of dimples which are uniform in size, which indicates that the fracture mechanism of the alloy is ductile fracture, so the double-wire wall has high strength and toughness.

## 4. Conclusions

In this study, Al–Cu–Sn alloy walls were manufactured using the WAAM process with single-wire and double-wire processes, respectively. The surface morphology, microstructure in the as-deposited and T6-treated states, and mechanical properties were investigated. The following conclusions are drawn:(1)At the same wire feeding speed, the heat input of the double-wire process was half that of the single wire, the surface quality of the double-wire wall improved, and accumulation efficiency also improved.(2)The number of pores in the double-wire wall was much smaller than that of the single-wire wall. The grains were mainly fine isometric crystals of uniform size, and the precipitated phases were diffusely distributed in the grain and on the grain boundary in the as-deposited state.(3)After T6 heat treatment, the *θ* phases of the double-wire wall dissolved into the matrix completely, and the number of *θ**’* phases was larger than the single-wire wall.(4)The tensile strength, yield strength, and elongation were 490 MPa, 420 MPa, and 12%, respectively, with no difference between the longitudinal and transverse. The fracture mode was a ductile fracture.

## Figures and Tables

**Figure 1 materials-13-00073-f001:**
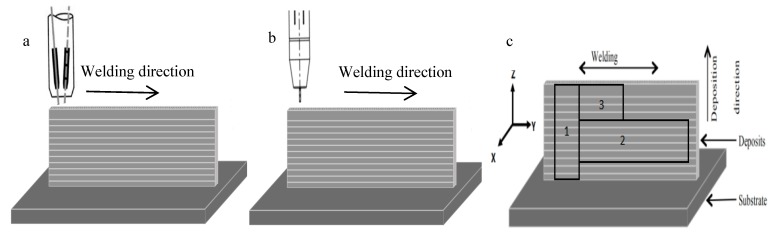
Schematic of the additive manufacturing system and sampling locations. (**a**) Schematic diagram of double-wire process; (**b**) schematic diagram of single-wire process; (**c**) sampling locations.

**Figure 2 materials-13-00073-f002:**
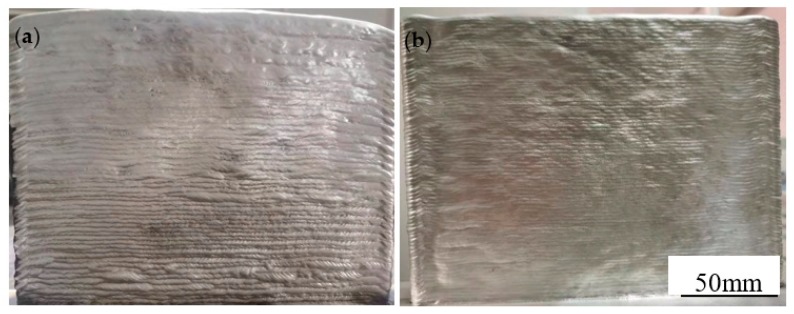
Surface morphology of single-wire and double-wire walls: (**a**) single-wire wall; (**b**) double-wire wall.

**Figure 3 materials-13-00073-f003:**
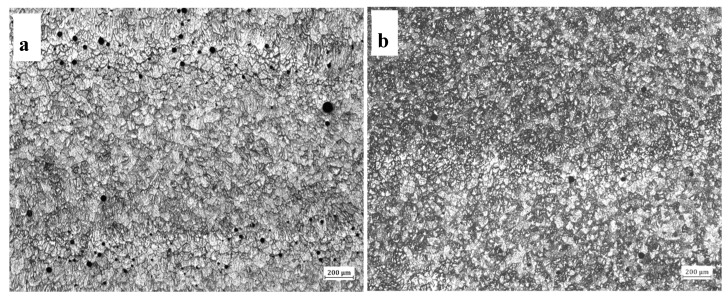
Porosity of single-wire and double-wire walls: (**a**) single-wire wall; (**b**) double-wire wall.

**Figure 4 materials-13-00073-f004:**
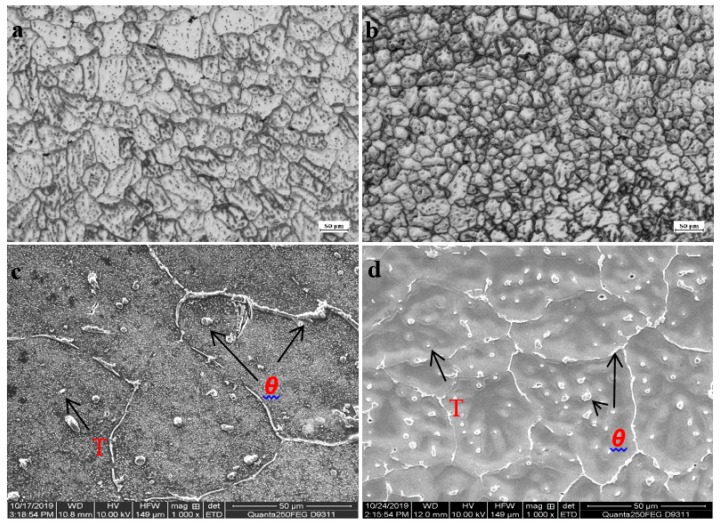
Metallographs of (**a**) single-wire wall and (**b**) double-wire wall; and SEM images of (**c**) single-wire wall and (**d**) double-wire wall in as-deposited state respectively.

**Figure 5 materials-13-00073-f005:**
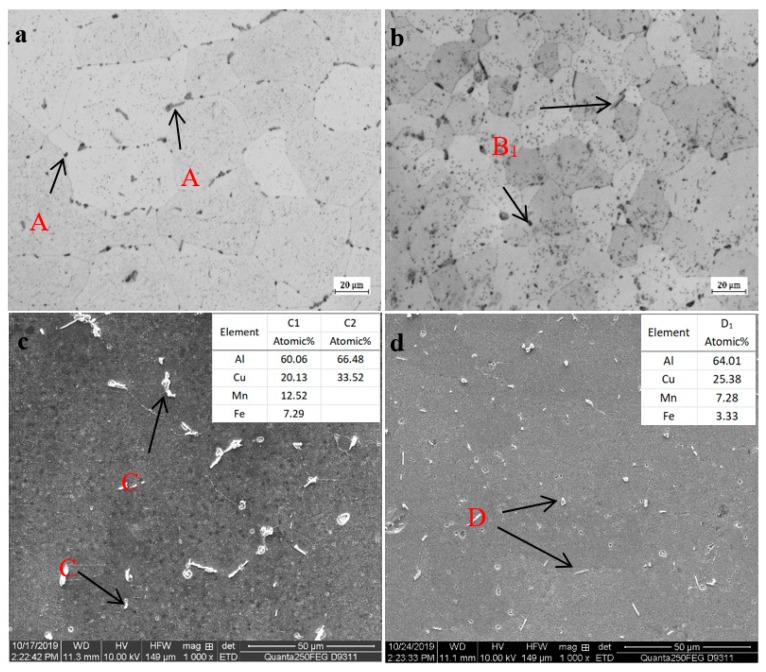
Metallographs of (**a**) single-wire wall and (**b**) double-wire wall; SEM and EDS of (**c**) the single-wire wall and (**d**) double-wire wall in T6 state.

**Figure 6 materials-13-00073-f006:**
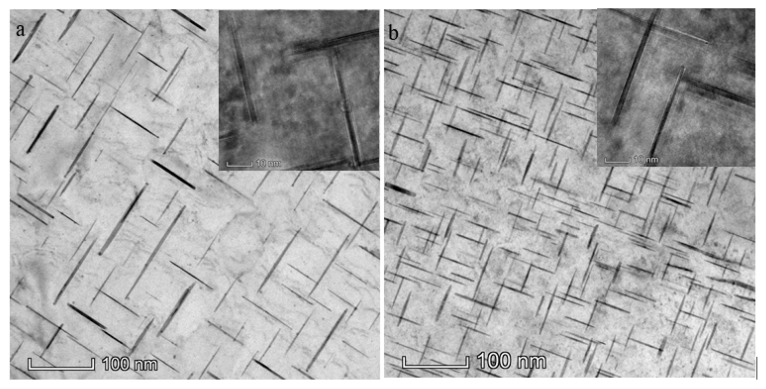
TEM images of strengthened *θ**’* phases at peak aging of (**a**) single-wire wall and (**b**) double-wire wall.

**Figure 7 materials-13-00073-f007:**
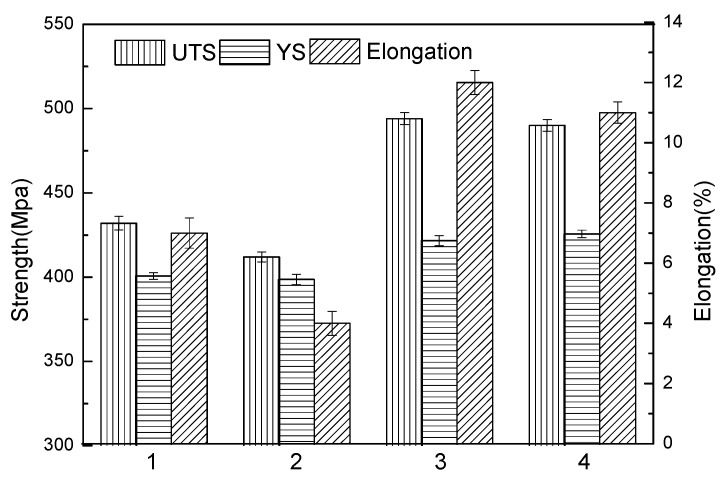
Mechanical properties of single-wire and double-wire wall: (**1**) transverse of single-wire; (**2**) longitudinal of single-wire; (**3**) transverse of double-wire; and (**4**) longitudinal of double-wire.

**Figure 8 materials-13-00073-f008:**
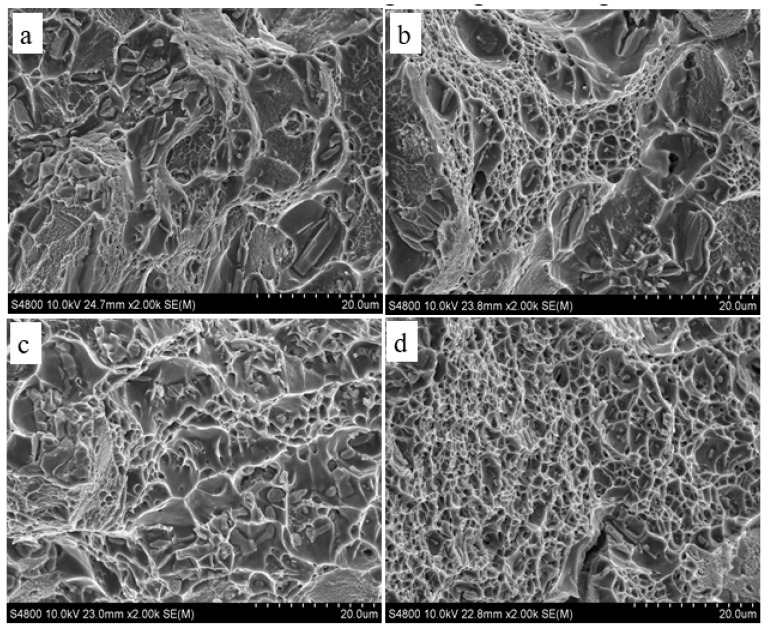
SEM images showing fracture morphology of single-wire and double-wire walls: (**a**) longitudinal of single-wire; (**b**) transverse of single-wire; (**c**) longitudinal of double-wire; and (**d**) transverse of double-wire.

**Table 1 materials-13-00073-t001:** Chemical composition of the raw material.

Element	Fe	Si	Mg	Cu	Mn	Ti	Sn	Zr	B	V
Content (wt %)	0.100	0.040	0.025	5.102	0.421	0.272	0.103	0.177	0.034	0.125

**Table 2 materials-13-00073-t002:** Print parameters.

Parameters	*I*/A	*U*/V	*ν_WFS_*/m·min^−1^	*ν_TS_*/m·min^−1^	Cool Time/s
Single wire	123	17.5	6	12	120
Double wire	56 + 56	14.6 + 14.6	3 + 3	12	120

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
