# Peer review of "Study on Microstructural and Mechanical Properties of an Al–Cu–Sn Alloy Wall Deposited by Double-Wire Arc Additive Manufacturing Process"

_materials, 2019, doi:10.3390/ma13010073_

Round 1
Reviewer 1 Report
All my comments are inside the attached paper.

Author Response
Response:Thank you for your comments. Inappropriateness of words and writing in the text has been modified. (1)The "CMT" is explained as follows: cold metal transfer; (2)The quality fraction is indicated in the chemical composition table of the raw materials; (3)Change "Dong Qing" to "Northeast Light Alloy Co., Ltd."; (4)Change "residual phases" to "unsolved phases".

Reviewer 2 Report
The paper entitled "Study on Microstructural and Mechanical Properties of an Al-Cu-Sn Alloy Wall Deposited by double-wires arc Additive Manufacturing Process” by Shuai et al. deals with the comparison of single and double wire arc additive manufacturing of Al-Cu-Sn parts. The process, microstructure and mechanical properties of the parts are analysed. Results are in the scope of the Materials journal. These are interesting, but after reading the paper, I have some comments about it:
GENERAL COMMENTS:
The introduction should be improved. It is not clear which is the main motivation of this paper. Because of this fact, the novelty of this research paper is not clear. Authors should notice that the main aspect analysed in this paper is the comparison of single and double arc wire additive manufacturing. Therefore, this should be highlighted in the introduction. Then, a proper review on this fact should be properly included and discussed. In the current state, authors have only added a collection of different works in additive manufacturing of Al alloys using wire arc methods. In the results section, it is discussed the pore formation taking into account the heat input and the nucleation rate. The results are plausible, but some clarifications are needed. First, the authors state that the heat input is a half in double-wire. This mainly is because they only take into account a half of the current in equation 1. However, the heat input in double wire should be larger because you are depositing both wires simultaneously. Probably, it is not so large as in single-wire, but also not a half. Please, clarify this in the manuscript.
PARTICULAR COMMENTS
(Page 1) Many acronyms (such as CMT, CMT+P+ADV) are used in the introduction without any prior introduction. Please, define the acronyms. (Page 1, Line 41) Please, replace Kg with kg along the manuscript. (Page 7, Fig. 7) Please, add error bars to Fig. 7.Author Response
(1)Comments:The introduction should be improved. It is not clear which is the main motivation of this paper. Because of this fact, the novelty of this research paper is not clear. Authors should notice that the main aspect analysed in this paper is the comparison of single and double arc wire additive manufacturing. Therefore, this should be highlighted in the introduction. Then, a proper review on this fact should be properly included and discussed. In the current state, authors have only added a collection of different works in additive manufacturing of Al alloys using wire arc methods. Response: Thank you for your valuable Suggestions. I have revised the introduction. In this paper, the purpose of using double-wire process is to reduce the heat input of accumulation. Under the premise of the same cladding efficiency, CMT-Tiwn can reduce the heat input significantly. (2)Comments: In the results section, it is discussed the pore formation taking into account the heat input and the nucleation rate. The results are plausible, but some clarifications are needed. First, the authors state that the heat input is a half in double-wire. This mainly is because they only take into account a half of the current in equation 1. However, the heat input in double wire should be larger because you are depositing both wires simultaneously. Probably, it is not so large as in single-wire, but also not a half. Please, clarify this in the manuscript. Response:there is an error in this paper, thank the reviewer for reminding. Under the same wire feed speed, single-wire feed speed is 6 m/min, the current is 123 A, voltage of 17.5 V, according to the formula: HI = η UI/vTS, HI (single-wire) = 143.5 J/mm . While double-wire wire feed speed is 3 + 3 m/min, the current is 56A, the voltage is14.6 V, the HI (double-wire) = 54.5 * 2 = 109 J/mm. The amount of heat input double-wire is less than single-wire, but does not half, has been modified in the article. The parameters of single-wire and double-wire are listed in table 2, and the calculation of heat input is given in article 3.2. (3)comments:(Page 1) Many acronyms (such as CMT, CMT+P+ADV) are used in the introduction without any prior introduction. Please, define the acronyms. (Page 1, Line 41) Please, replace Kg with kg along the manuscript. (Page 7, Fig. 7) Please, add error bars to Fig. 7. Response:The "CMT" is explained as follows: cold metal transfer;CMT+P+ADV has been delete due to revision of the introduction;Error bars have been added in Fig.7.

Reviewer 3 Report
In general, the paper is well-written and presents interesting results with a decent discussion, which are supported by the experimental data.
I have a few minor comments:
- adding a schematic representation of a single-wire and double-wire processes to the Experimental section would be helpful.
I think the following sentence in 3.4 section and Conclusions needs to be restructured because now it gives a wrong idea that, for example, the maximum tensile strength is 78 MPa with the elongation of 175%.
"Compared with single-wire wall, the mechanical properties of the double-wire wall are greatly improved, including tensile strength of 62 MPa, yield strength of 21Mpa, and elongation of 70% in the transverse, and tensile strength of 78Mpa, yield strength of 27Mpa, and elongation of 175% in longitudinal."
Author Response
adding a schematic representation of a single-wire and double-wire processes to the Experimental section would be helpful.
Response:The schematic diagram of double wire and single wire will be useful for the explanation of experiment, which has been added in this paper.
I think the following sentence in 3.4 section and Conclusions needs to be restructured because now it gives a wrong idea that, for example, the maximum tensile strength is 78 MPa with the elongation of 175%.
"Compared with single-wire wall, the mechanical properties of the double-wire wall are greatly improved, including tensile strength of 62 MPa, yield strength of 21Mpa, and elongation of 70% in the transverse, and tensile strength of 78Mpa, yield strength of 27Mpa, and elongation of 175% in longitudinal."
Response:Sentence in 3.4 section have been modified to “Compared with single-wire wall, the mechanical properties of the double-wire wall are greatly improved, tensile strength: 490Mpa, yield strength: 420Mpa, elongation: 12%,the mechanical properties of horizonal direction and vertical direction are consistent.”
Sentence in conclusions have been modified to“The tensile strength, yield strength, and elongation are 490 MPa, 420 MPa, and 12%, respectively with no difference between the longitudinal and transverse, and the fracture mode is ductile fracture.”
